# COVID-19, Race, and Crime: An Early Look at Racial Disparities in U.S. Arrest Data throughout the COVID-19 Pandemic

**Calvin Proffit * and Ben Feldmeyer**

School of Criminal Justice, Main Campus, University of Cincinnati, Cincinnati, OH 45221, USA; feldmebn@ucmail.uc.edu
* Correspondence: profficl@mail.uc.edu

**Abstract:** Background: This study explores how arrests changed in response to the COVID-19 pandemic across race. Daily life changed for everyone across the country with the onset of the pandemic, and early works have shown that crime changed in this period. Method: Official arrest data were pulled from the Indiana State Police database for several violent and property crimes covering 26 counties. Data were gathered from 2017 to 2021 for a comparison of pre-COVID-19 versus after the onset of COVID-19 (2020–2021). An OLS regression was run to assess differences in these patterns of arrests across Black and White populations. Results: This analysis finds that Black homicide, White homicide, and total Black violent crime arrests were significantly related to COVID-19 measures after controlling for other variables. The COVID-19 measures indicate that these crimes saw an increase in arrest after the onset of the pandemic and that these effects may not have been identical across race. Conclusions: The COVID-19 pandemic was linked to crime across race in the state of Indiana. Moving forward, it is important to uncover how crime changed across race in other locales and exactly what mechanisms may have driven these changes.

**Keywords:** COVID-19; crime; arrests; race; pandemic; violence





## 1. Introduction

The COVID-19 pandemic affected nearly everyone in the United States in one way or another. By 2023, the United States had experienced more than 1.1 million verified deaths from the COVID-19 virus [1]. More than two years after the virus first made its way to the United States, societal changes resulting from the pandemic persist and continue to accumulate. Many people started to work from home, attend online school, or social distance in public. In March 2020, quarantine and lockdown orders went into effect, forcing large numbers of businesses to shut down, many of which would never reopen. Initially, the lockdown was intended to last for only two weeks, but it eventually turned into months and years, with COVID-19's effects persisting for years after the virus subsided. By October 2021, the U.S. job market had 4.2 million fewer jobs than before the pandemic [2], and roughly 25% of adults were having difficulty covering household expenses. Forced isolation affected mental health as fear of the disease spread across the nation [3]. Lifestyles changed, and daily activities became radically different.

Given these widespread social changes, there is reason to expect that crime also fluctuated with these shifts in behaviors, lifestyles, and routines. Reports indicate that during the pandemic, property crimes declined, but homicide saw a 35% increase nationally during the first 6 months of 2020 [4]. The full scope of these changes in crime is still unknown, but some initial research has provided insight into how crime generally shifted in response to the pandemic. Police practices changed, criminal opportunities shifted, and the economy suffered greatly [5]. The police started to make fewer arrests and fewer traffic stops, and some crimes were punished less severely [6,7]. Families were home more often, which might suggest a potential decline in home invasions [4,5]. At the same time, when

couples are confined to the home, this could lead to a rise in cases of domestic violence [8]. However, questions remain about the ways in which these and other crime patterns may have changed throughout the pandemic. Early reports have shown a combination of spikes and stability (depending on the location) for homicide and assaults, but more research is needed to develop a better understanding of these changes in violence as well as the effects of the pandemic on other crimes [5].

Due to the recent nature of the pandemic, and the fact that its impact is still unfolding across U.S. communities, many questions remain about its effects on crime. In addition, data allowing systematic assessments of COVID-19–crime relationships have only recently been released. Notably, the FBI began to phase out the Uniform Crime Report (UCR) and transferred to the National Incident-Based Reporting System (NIBRS) precisely as the COVID-19 pandemic occurred. Data for this period are just now becoming available for analysis, but due to the transition of databases, assessing long-term nationwide effects of COVID-19 on crime has been difficult. A few select studies have offered initial assessments of how COVID-19 impacted crime, but research on the topic remains limited. Among the studies that have examined COVID-19's relationship with crime, evidence is mixed (described in more detail below). Some show decreases in several crime categories throughout the pandemic but stable or increasing rates for other offenses, such as violent crime [5,8]. In addition, there is almost no research examining COVID-19's effects on crime *across race*, which is the focus of the current study. How did COVID-19 impact Black arrests specifically, and were the effects similar to those seen for Whites? Were there greater disparities in arrests between White and Black citizens during this period? Did COVID-19 affect arrests for all race/ethnic groups in "racially invariant" ways? Or were the effects of COVID-19 on crime felt more by some race groups than others, and did this differ by offense type?

As we review below, there are competing arguments about how and why race could matter in COVID-19–crime relationships. On one hand, there is reason to believe that COVID-19's impact on crime may have been similar across race. The racial invariance theory suggests that the causes of crime are the same across race [9]. Based on this perspective, the relationships between COVID-19 and crime should be similar for all racial groups. Afterall, the pandemic affected all Americans and all race/ethnic groups. Black and White populations alike experienced social distancing, social isolation, and infection from the disease, and no group was immune to the social or health impacts of the pandemic. Thus, the effects of COVID-19 on crime may be similar for both Black and White populations. On the other hand, several arguments drawn from strain theory and from research on race and disadvantage suggest that COVID-19's effects on crime may not be equal across race. COVID-19 caused strains and disruptions for everyone, but it likely disrupted life more for those who had fewer resources. Due to the already heightened levels of disadvantage among Black populations and the added challenges imposed by COVID-19, there is reason to believe that the pandemic's impact on crime may have been stronger for minority groups than for Whites [10–12].

This study provides an exploratory analysis examining macro-level relationships between COVID-19, race, and crime to see how White and Black county-level arrest rates changed from pre-COVID-19 (2017–2019) compared to COVID-19 onset years (2020–2021). This paper proceeds as follows. We begin by reviewing the existing literature to examine COVID-19's broader impact on crime and discuss theoretical reasons for which COVID-19 may have impacted crime rates (overall and across race). We then describe the data and methods used in the current analysis. Specifically, this analysis uses official police data from Indiana to examine Black and White arrests for homicide, aggravated assault, burglary, and a broader "violent crime" measure (includes abuse, battery, domestic battery, domestic violence, homicide, intimidation, robbery, strangulation, and other) covering the 2017 to 2021 period. We then present multivariate analyses using OLS regression to assess whether the relationships between COVID-19 and arrests differ for White and Black populations (net of county-level controls for demographic and social context). Last, we conclude by discussing the implications of these findings and directions for future research.

## 2. Prior Research on COVID-19 and Crime

The pandemic was a unique time in history. Public health crises of this magnitude are rare, and their impact is often not fully understood until years later. COVID-19's impact on daily life was dramatic for everyone, and as discussed below, several criminological theories suggest that crime may have changed due to these shifts and restructuring of society during the pandemic.

Lifestyles shifted dramatically during the pandemic, which, according to opportunity and routine activities perspectives, likely changed opportunities for crime [7]. What used to be suitable targets for crime were no longer available. For instance, according to routine activities theory, homes may have become less suitable for burglary due to large segments of the population being confined to their homes [13]. In support of this position, national crime data showed a 52.5% decrease in property crimes after the SAH orders were enacted, with a 20% reduction in residential burglary [14]. In addition, reports show that these reductions in burglary occurred across major U.S. cities such as Denver, Houston, New York, and Sacramento [4]. However, while these cities saw reductions in residential burglary, other crimes increased. Vehicle larcenies increased by 9% [4]. Commercial burglaries rose in these cities as well, perhaps due to people spending more time at home, thereby reducing guardianship of commercial establishments. Notably, routine activities and opportunity perspectives also suggest that the pandemic could have caused shifts in violence. On the one hand, violent street crimes may have decreased because people had much more limited contact due to social isolation and thus reduced opportunities for violent interactions. On the other hand, this could also have led to increases in opportunity for some forms of violence, such as domestic assaults. In addition, with society's shift indoors, social guardianship declined, leaving room for an increase in suitable targets for violence and victimization on the street.

Strain theory might also suggest an increase in violence stemming from the pandemic. During the early phases of COVID-19, unemployment rose, family dynamics changed for many, and tensions grew [3,7]. Mainstream goals were blocked for many as jobs were lost, education was put on hold, and stay-at-home orders increased social isolation and stress [10]. Negative stimuli were introduced into many people's lives as illness ran rampant across the country and loved ones died from the virus. An increase in strain may have created shorter fuses and greater interpersonal violence as lives were changing dramatically in this time [15]. These blocked goals and negative stimuli may have contributed to more violence according to strain theories, which could be reflected in potential increases in homicide, assault, and domestic violence [16]. In line with these arguments, reports indicate that across 51 cities, there was a 35% increase in homicide rates on average from 2019 to 2020 [4]. For example, Detroit saw a 68% increase in homicide, and Houston saw a 19% increase in aggravated assaults during the first six months of 2020 compared to a similar timeframe in 2019 [4]. During this same period, assaults with a firearm saw a 10% increase across 21 cities [4], suggesting that at least some forms of violence may have escalated in response to the pandemic.

### 2.1. Empirical Research on COVID-19 and Crime

There is little research thus far examining the relationships between COVID-19 and crime. However, there are a few noteworthy studies that have begun this work and explored how crime changed across the United States during the pandemic. An early empirical study by David S. Abrams provides an initial look at how crime changed across 25 major U.S. cities during the pandemic [5]. This study suggests that it was not until the stay-at-home (SAH) orders began that crime saw a dramatic change. After the SAH orders went into effect, there was a significant initial drop in official crime incidents, including a 23.3% decline in overall crime [5]. Abrams notes that police began to focus less on certain crimes, which may have contributed to a 65% decrease in drug offense arrests seen during this period [5]. Across the 25 major U.S. cities examined in the study, property crimes fell significantly, except for non-residential burglary and car theft. The author suggests that

this is likely due to increased opportunities for burglarizing commercial properties that were closed [5].

Violent crime provided a different picture. Abrams notes that while some violent crime decreased, homicides and shootings did not [5]. Some cities reported higher levels of homicides than in previous years, which continued into 2022 based on arrest data [4]. Abrams notes that the variation across cities was substantial [5]. This may be due to reporting issues during the pandemic, changes in police practices, reduced enforcement for some offenses (i.e., drug offenses), as well as the shift to the NIBRS. There is a chance that these reporting issues had an effect. Regardless of the precise source, Abrams's work illustrates clear initial changes in arrest patterns accompanying the onset of the pandemic. Yet, this work does not include demographic comparisons to assess whether these patterns differ across race/ethnicity, which is the focus of the current study.

In a similar study, Ashby examined 16 cities across the United States to identify initial changes in crime during the pandemic [8]. Ashby specifically examined serious assaults in public, serious assaults in residences, residential burglaries, non-residential burglaries, theft of vehicles, and theft from vehicles [8]. This study was unique in that it included a forecast of predictive crime rates versus actual crime records. The findings of the study suggested that crime changes were not uniform across cities, but a notable limitation was the nature of the predictive models used since they rely on the assumption that no changes in crime occurred from the previous years [8]. One finding showed that there were no significant changes in serious assaults in public (contrary to Abrams's work in 2021). Cities showed decreases in residential burglaries, but not commercial property burglaries, as seen in other work. Motor vehicle thefts varied between cities, which was shown in Abrams's work as well. While residential assaults showed a decrease, it is important to note that the reporting of these crimes likely dropped significantly, which may have contributed to these trends [8].

While the two studies described above examined official crime incidents across cities, several other studies examined crime by measuring calls for service. Boman and Gallupe compared 2019 and 2020 data, looking to see if calls for service changed during the pandemic [17]. They found that calls for service declined dramatically (roughly from 12 to 25%) after the onset of the SAH orders [17]. They note that it is unclear whether this reflects reduced reporting or decreases in actual levels of crime due to changes in day-to-day life (or both). For example, they note that commercial properties could have experienced burglaries that went unreported for months while they were temporarily closed. However, not all calls for service decreased according to their study. Minor crimes decreased, which could be explained by the fact that many low-level crimes are committed in groups, and peer groups were less likely to be meeting due to the SAH orders [17]. However, serious violent crime, which is often perpetrated by a single individual, remained steady and did not change noticeably within the cities examined in their analysis. Philadelphia and Chicago both reported consistent homicide rates between 2019 and 2020, suggesting that the pandemic had no notable effect on homicide in these locales [17]. Unlike other studies, Boman and Gallupe also found an increase in interpersonal violence (IPV) incidents in Philadelphia—specifically, a 33% increase between 2019 and 2020. The police chief reported that it was likely due to the SAH orders and increases in alcohol consumption (seen through sale spikes).

A recent Special Issue of *Criminology & Public Policy* focusing specifically on COVID-19's effects on crime and punishment offers several additional studies examining connections between COVID-19 and crime. One study in this issue by Richards et al. examined how DV calls changed with the onset of the SAH orders [18]. Their analysis indicated that compared to 2018 and 2019, DV-related calls increased (net total of 2700) across some cities [18]. However, they note the stark differences seen across jurisdictions in terms of calls to police. Another study in this issue reported that gang violence was unaffected by COVID-19 and the SAH orders [19]. The author's analyses showed that there were no

significant changes in gang-related assaults, gun violence, robberies, or general violence with the onset of the pandemic.

Overall, the initial wave of research examining how COVID-19 impacted crime has shown some consistent patterns. This line of work indicates that crime rates generally dropped, especially for property and minor offenses. However, serious violent crimes, such as homicides, shootings, and aggravated assaults, may have remained constant or increased during the pandemic [3,4,8,17]. A variety of explanations about how COVID-19 impacted crime have been provided in prior research. Whether it is because of social changes, strain, SAH orders, policing and crime reporting changes, or opportunity shifts, there are several plausible reasons why COVID-19 may have had an impact on crime. However, work examining these relationships to date has largely focused on total or overall effects of COVID-19 on crime, with almost no research examining these relationships across race/ethnicity, which we turn to below.

### 2.2. Race/Ethnicity, COVID-19, and Crime

Although there has been growing attention on COVID-19's effects on crime, very few studies to date have incorporated race into their analysis of COVID-19–crime relationships. One study by Dunbar and Jones examined why Black residents may be at a higher risk for police interaction when not abiding by public health guidelines [17]. They found that data on arrests for violations of public health orders showed a disproportionate level of Black arrests [20]. Looking specifically at New York, where 24% of the population is Black, roughly 68% of social distance violations for court summons were for Black residents, with similar statistics shown in Chicago. Ohio arrest data showed a disproportionate level of Black residents involved in SAH order violation arrests as well [20]. Notably, this pattern of disproportionate arrests may extend to other offenses, not just SAH orders, and could lead to significant race differences in arrests during the pandemic.

Similarly, reports from the Los Angeles Police Department showed some disparities in crime patterns across communities of color throughout the pandemic in an analysis of 27 cities [4]. Specifically, these data showed that while violent crime declined in majority-White communities during the SAH orders, rates of violence remained stable in majority-Black communities. Even after the SAH orders expired, violence levels remained on the rise in majority-Black communities but not in majority-White communities [4], suggesting potential unequal impacts of the pandemic across race. However, outside of these studies, research has given almost no attention to the ways in which COVID-19's impact on crime compares across race.

### 2.3. Theoretical Arguments about Race, COVID-19, and Crime

There are several theoretical reasons why we might expect either similarities or differences in the impact of COVID-19 on arrests across race, especially for Black and White comparisons. As noted above, strain theories might suggest increasing rates of violent crime in response to COVID-19. Employment, family structure, and lifestyles changed dramatically during the pandemic, which may have contributed to greater pressures toward crime (especially for violent offenses). However, this may have impacted some race/ethnic groups more than others. Specifically, those with fewer resources may have experienced the effects of these strains more dramatically. This may especially be the case for minority populations living in disadvantaged areas. As prior research has shown, even the most disadvantaged White neighborhoods are typically better off than nearly half of all Black neighborhoods [9,12,21]. As such, the strains from COVID-19 may have had even greater, compounding effects on Black populations that were already disadvantaged before the pandemic. Individuals who could not work from home due to the nature of their job may have been without employment for an extended time or laid off due to businesses closing. Youth who attended schools that did not have access to funding for personal laptops likely faced more challenges than wealthier school districts. Unemployment was flourishing, adolescents were struggling to attend school, and family resources became stretched—*but*

*especially for minority groups that were already disadvantaged* [2,4,22]. Thus, COVID-19 may have generated greater strains and pressures toward crime, especially among minority populations who often had fewer resources to begin with. In sum, strain perspectives and research on concentrated disadvantage would suggest that some racial groups may have been less impacted by the onset and disruptions of COVID-19 (e.g., upper income White populations), whereas others may have experienced even greater, compounding levels of strain that added to the deprivation and crime-generating sources of disadvantage they already faced.

In contrast, the "racial invariance" theory would suggest that the effects of COVID-19 on crime may be similar across race/ethnicity. This theory argues that racial differences in crime are not directly due to race per se, but rather are due to group differences in social and economic conditions [9]. Rather than focusing on cultural differences or looking for different causes of crime across racial groups, this theory suggests that the sources of crime for Black and White populations (and other groups) are generally the same. That is, the causes of crime are "invariant" for all race/ethnic groups according to this perspective, perhaps including the effects of the pandemic [9,21]. Racial invariance would suggest that if the pandemic impacted crime, it would likely exert similar effects on crime across races. In sum, this perspective would suggest that there is nothing that would inherently make COVID-19 a greater crime-generating factor for one group versus another. It may have changed social life and strains, but these broad changes occurred for everyone. All groups experienced some form of strain, such as the stay-at-home orders and social distancing, regardless of race or ethnicity. While these changes may have varied in degrees, there was nothing inherently race-based or fundamentally different "in kind" about these experiences that were explicitly linked to race. This is not to say that White and Black communities experienced exactly the same consequences from the pandemic. Social changes were felt differently across groups and communities [2], but we may not see this change reflected in COVID-19's effects on crime net of other factors. As such, we might expect COVID-19 to have had similar or "racially invariant" effects on crime for both White and Black populations after accounting for other social and economic conditions that each group experienced [9,23].

In light of these alternative arguments, this study will specifically compare Black and White arrest patterns for burglary and several violent offenses to identify race differences (or similarities) in COVID-19–crime relationships. Given the competing theoretical arguments described above, it is unclear exactly whether and how COVID-19 may have shaped arrest rates across race. Thus, the current study seeks to provide an early, exploratory assessment of the ways in which arrests changed during the pandemic across race.

## 3. Materials and Methods

### 3.1. Data

Due to the recent nature of the COVID-19 pandemic, crime data are not yet widely available for pre- and post-COVID-19 comparisons in many commonly used crime databases. The FBI's Uniform Crime Report has limited data availability after 2019, when it began to fully transition to the National Incident-Based Reporting System (NIBRS). This shift disrupts the ability to look at arrest trends covering the 2017 to 2021 period and complicates the use of UCR data for examining crime data covering the onset and duration of the pandemic. Several large states, such as New York and California, have some data available. However, due to funding and other barriers from COVID-19, crime data are not available to the public or have not yet been fully released for 2020 to 2021 in most states. In contrast, the Indiana State Police offer crime data covering 2017 through 2021 via the Indiana Data Hub, data which were used in this study. Data for 2022 were excluded because crime reporting for this year was not complete at the time of analysis. However, the early impact of COVID-19 can be assessed using data from 2020 and 2021 (versus earlier years) due to the overwhelming impact of the pandemic in these years. The Indiana State Police website has an interactive map examining county data for all crime types defined in their state.

For this study, we focused exclusively on arrests and excluded data from later stages of criminal justice processing (such as disposition or incarceration).

As a unit of analysis, twenty-six counties were examined, which include those with available measures for all control variables[1]. Each county had racial, sex, and age breakdowns for arrests, but only racial breakdowns were examined for the following offenses: "violent crime" arrests, homicide arrests, assault arrests, and burglary arrests. These crime categories were chosen due to their seriousness and less susceptibility to reporting biases, as well as their focus in prior literature on COVID-19 and crime [24–26]. As the reader may recall, violent crimes generally remained stable or increased during COVID-19 based on prior literature, hence our focus on violent crime arrests and homicide here. However, we also examine burglary to provide some analysis of property arrests and to assess whether arrests for burglary in Indiana declined, as shown in some other state analyses [5,8]. Data on arrests and control variables were gathered from 2017 to 2021.

### 3.2. Dependent Variable

Using Indiana police data from 2017 to 2021, information on arrests was gathered for four offense types. The dependent variables examined here are measured as arrest rates per 100,000 for a broad "violent crime" category, as well as for homicide, aggravated assault, and burglary. In the Indiana police data, the "violent crime" category includes the following offenses: abuse, battery, domestic battery, domestic violence, homicide, intimidation, robbery, strangulation, and other. Although the violent crime category includes assaults and homicide, both of these offenses were also examined separately. Indiana defines assault (battery) as when "a person who knowingly or intentionally touches another person in a rude, insolent, or angry manner; or places any bodily fluid or waste on another person" [27]. Burglary was included as the only property crime due to its prevalence in COVID-19–crime literature and to examine if prior trends for this offense in other states (which often showed declines) are mirrored in Indiana data.

### 3.3. Independent Variables

The key measure of interest in this analysis is the COVID-19 onset measure. This variable is measured using a binary code to account for the onset of COVID-19 in 2020. All years prior were coded as 0 for the pre-COVID-19 period, and dates for 2020 and 2021 were coded as 1 to capture years in which the COVID-19 pandemic was fully present. However, we also examined the county-level COVID-19 death rate (COVID-19 deaths/100,000 residents) as an alternative measure of COVID-19's impact on each county, which is discussed in the supplemental analysis section later in the paper. We obtained these data from the CDC COVID-19 death statistics [1].

Control variables were gathered using Social Explorer's American Community Survey (ACS) 1-year estimates for 2017 through 2021. These variables were gathered across the twenty-six counties examined here for which all measures were available, and they account for county social and economic conditions commonly examined in macro-level research on crime [11,21,28,29]. Educational attainment was controlled using the "some college" variable, which measured the percent of the county who obtained at least 1 year of college credits. Residential mobility was measured as the percentage of the population that had lived in their house for more than one year. Disadvantage was controlled with two measures capturing (1) the percentage of the county population that was unemployed and (2) the average county household income. The percentage of the county population without health insurance was also controlled as a potentially important variable associated with COVID-19 illness[2].

### 3.4. Analytic Approach

For this analysis, an ordinary least squares regression (OLS) was used due to the linear distribution of the outcome variables (arrest rates) and the desire to examine effects of COVID-19 onset, net of disadvantage, educational attainment, residential mobility, and

health insurance coverage. As seen in the following models, the COVID-19 measure, which is the binary coding of year (1 = COVID-19 year, 0 = pre-COVID-19 year), is in the last row. This analysis allows for isolation of this variable from the other five controls to determine the significance of COVID-19 net of other variables. Significance was measured at the $p < 0.01$ and $p < 0.05$ levels. OLS models were run overall and for each race group (White and Black) and across each offense type (total violent crime, homicide, assault, and burglary). This resulted in 8 separate OLS models, the results of which are described below. In the following section, we first provide descriptive plots of crime trends during the study period to offer a picture of shifts in arrests for Indiana counties (overall and by race). We then present results of the OLS models, which show the relationships between COVID-19 and crime for both White and Black arrests, net of controls[3].

## 4. Results

Focusing first on the descriptive picture of Indiana arrests between 2017 and 2021, the total arrest count is $n = 677,386$ across the 26 counties examined. Table 1 provides a breakdown of arrests across race for each offense. Several trends can be seen in the data. First, total arrest counts generally increased across the state of Indiana for both Black and White populations, which is reflected in Figure 1. Black and White violent crime arrests remained stable from 2017 to 2019, with a small increase beginning in 2020, which is reflected in Figure 2. Interestingly, Black and White homicide saw an increase in 2019 that continued to increase during the COVID-19 pandemic, which is seen in Figure 3. All counts seen in the figures below are reflective of the analytical sample.

**Table 1.** Descriptives across offence arrests (2017–2021).

| | 2017 | | 2018 | | 2019 | | 2020 | | 2021 | |
|---|---|---|---|---|---|---|---|---|---|---|
| | Freq. | % | Freq. | % | Freq. | % | Freq. | % | Freq. | % |
| Total Arrests | 110,285 | | 112,681 | | 149,720 | | 152,223 | | 152,477 | |
| Black | 35,592 | 32% | 37,215 | 33% | 46,510 | 31% | 47,494 | 31% | 47,494 | 31% |
| White | 72,992 | 66% | 73,544 | 65% | 100,197 | 67% | 103,346 | 68% | 103,893 | 68% |
| Violent Arrests | 19,873 | | 20,107 | | 20,194 | | 20,307 | | 21,224 | |
| Black | 8117 | 41% | 8126 | 40% | 8138 | 40% | 8220 | 40% | 8537 | 40% |
| White | 11,453 | 58% | 11,482 | 57% | 11,709 | 58% | 11,852 | 58% | 12,438 | 59% |
| Homicide Arrests | 307 | | 328 | | 331 | | 388 | | 431 | |
| Black | 194 | 63% | 204 | 62% | 218 | 66% | 241 | 62% | 268 | 62% |
| White | 96 | 31% | 107 | 33% | 134 | 40% | 140 | 36% | 154 | 36% |
| Assault Arrests | 7522 | | 7848 | | 8814 | | 9227 | | 9571 | |
| Black | 3252 | 43% | 3374 | 43% | 3750 | 43% | 4044 | 44% | 4066 | 42% |
| White | 4158 | 55% | 4367 | 56% | 4988 | 57% | 5058 | 55% | 5325 | 56% |
| Burglary Arrests | 1236 | | 1402 | | 1569 | | 1753 | | 1793 | |
| Black | 361 | 29% | 390 | 28% | 426 | 27% | 497 | 28% | 609 | 34% |
| White | 860 | 70% | 996 | 71% | 1126 | 72% | 1170 | 67% | 1232 | 69% |

We now turn to the OLS multivariate models to assess the relationships between COVID-19 and arrests, net of controls (Table 2). Focusing first on our key variable of interest, Table 2 of the OLS regression shows several noteworthy relationships between the COVID-19 year variable and arrest rates. COVID-19 year had a positive and significant effect on Black violent crime arrests at the 0.05 level, indicating that Black arrest rates for violence were higher in COVID-19 years (compared to pre-COVID-19 years). COVID-19 year also had positive and significant effects on Black homicide arrests at the 0.05 level and White homicide arrests at the 0.01 level. This indicates that violent crime, especially homicides, was higher after the onset of COVID-19, net of other factors. In addition, the current results show some differences across race. Specifically, there was no significant effect for COVID-19 year on arrests for the White violent crime category, but this effect was positive and significant for the Black population. Significance for several types of Black violent arrests, but only for White homicide arrests, suggests that the COVID-19 effect

on crime, and especially violence, was more consistent and may have mattered more for Black arrests.

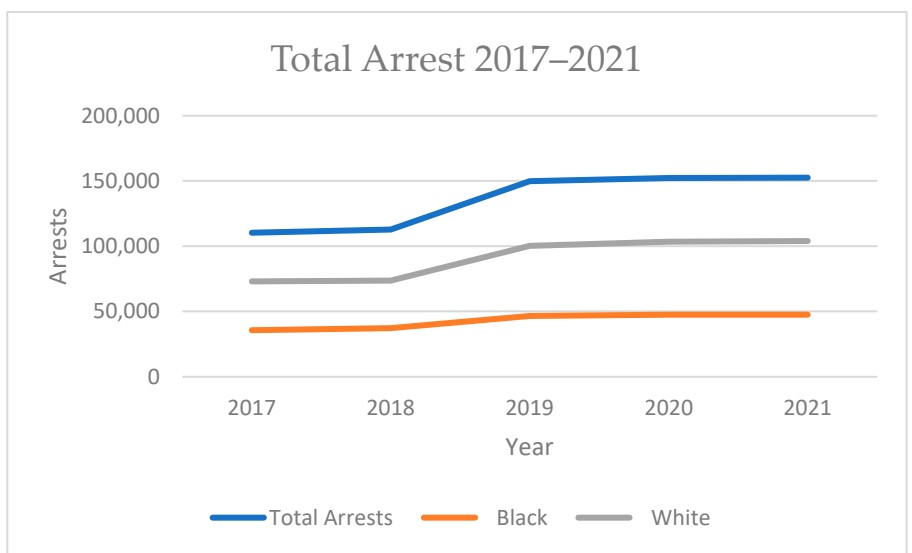

**Figure 1.** Total Indiana arrest counts for White and Black populations, 2017–2021.

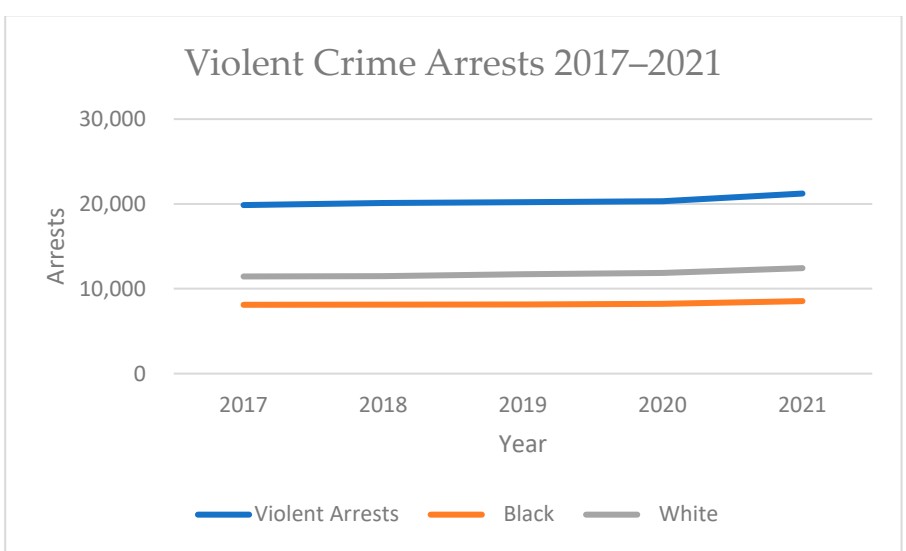

**Figure 2.** Total Indiana violent crime arrest counts for White and Black populations, 2017–2021.

Turning to the control variables, across all models and all offenses examined, average household income had a significant negative effect on arrests. Unemployment had a positive effect on arrests and was significant for White violent arrests and Black burglary arrests at the 0.05 level and for Black and White assault and White burglary at the 0.01 level. The significant effect of unemployment indicates that violent crime arrests were consistently higher in counties with greater unemployment. Notably, these effects of unemployment and income were linked to arrests for both groups in similar ways for the most part. The education measure had mixed effects across groups. Educational attainment was significantly related to arrests for Black violent crime, White homicide, and Black assault at the 0.05 level, and these effects were significant at the 0.01 level for Black homicide and White assault. However, the effect of education varied in direction. Specifically, Table 1 shows that higher education was linked to lower arrest rates for Black and White assault and White burglary. In contrast, education was associated with higher arrest rates for Black violent crime and for Black and White homicide. Lower levels of health insurance coverage

were positively related to arrests for Black homicide and Black assault at the 0.01 level but had almost no effects in the White models, with one exception. This finding suggests there may be some differences in the predictors of arrest across race. These models highlight that many structural factors were related to changes in arrests during COVID-19, and some but not all of the structural conditions affected both groups. However, homicide for Blacks and Whites and violent crime arrests for Blacks were significantly related to our COVID-19 measure, suggesting some difference across race in the impact of COVID-19 and other structural conditions on Indiana arrests.

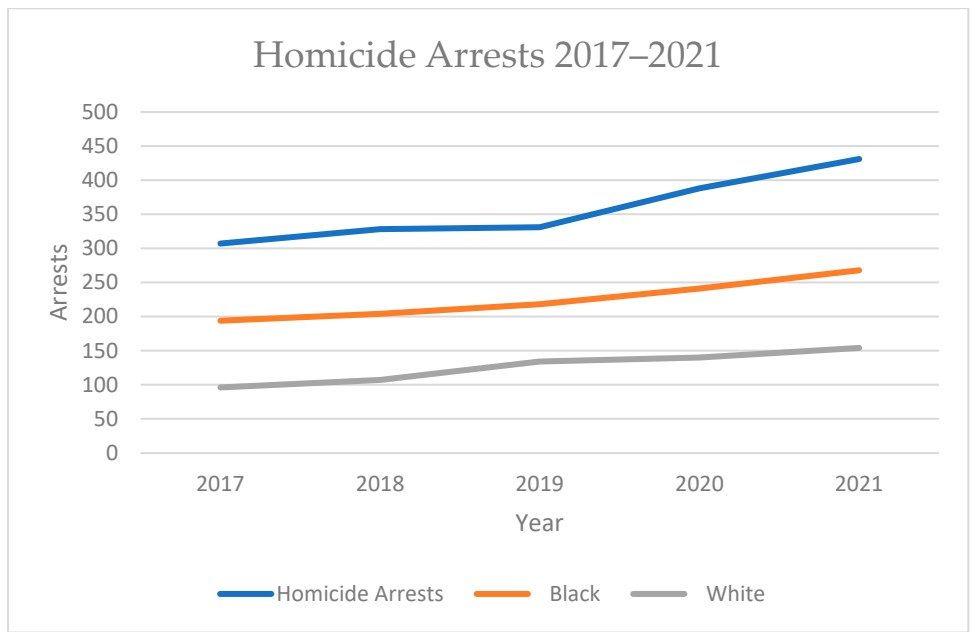

**Figure 3.** Total Indiana homicide arrest counts for White and Black populations, 2017–2021.

*Supplemental Analysis*

To further exhaust the data, we conducted a supplemental analysis using the mortality rate from COVID-19 for each county (Table 3). This was included to see if counties with higher COVID-19 mortality rates had increased arrest rates (rather than relying on bivariate year measures as used in the previous models). Theoretically, communities that had higher mortality rates would likely have faced greater impacts from the pandemic. The mortality rate provides an alternative measure of COVID-19's impact to offer a robustness check for the prior analysis. In this analysis, the mortality rate measure replaced the prior COVID-19 binary year measure. Using the alternative COVID-19 death rate measure, our models produced similar results. As seen in our earlier models, the COVID-19 death rate had a positive and statistically significant effect on White homicide arrest rates, Black homicide arrest rates, and total Black violent crime arrest rates. This reproduction of similar findings across two COVID-19 variables reaffirms the results of the prior analysis and suggests that the pandemic was tied to arrests for these groups and arrest categories.

We also ran negative binomial models as a final robustness check because there was skewness in county crime rate measures (i.e., homicide). Thus, count-based models are often used to adjust for this skewness. We reran all models shown in the main analysis using negative binomial models which produced similar findings to those presented here. Specifically, the supplemental models showed significant COVID-19 effects at the 0.05 level for Black and White homicide arrests (the results are available upon request).

**Table 2.** Ordinary least squares regression of arrest rates for White and Black populations in Indiana.

| | Black | | | | White | | | |
|---|---|---|---|---|---|---|---|---|
| | **Violence** | **Homicide** | **Assault** | **Burglary** | **Violence** | **Homicide** | **Assault** | **Burglary** |
| | *b* (SE) | *b* (SE) | *b* (SE) | *b* (SE) | *b* (SE) | *b* (SE) | *b* (SE) | *b* (SE) |
| Some College | 0.002 * | 0.000 *** | −0.001 ** | 0.000 | 0.001 | $7.149 \times 10^5$ ** | −0.002 *** | −0.001 * |
| | (0.001) | (0.000) | (0.000) | (0.000) | (0.002) | (0.000) | (0.001) | (0.000) |
| Unemployment | −0.002 ** | 0.000 | 0.006 *** | 0.001 ** | 0.018 ** | 0.000 | 0.013 *** | 0.005 *** |
| | (0.004) | (0.000) | (0.002) | (0.000) | (0.008) | (0.000) | (0.003) | (0.001) |
| Average Household Income (measured in $10,000 units) | −0.002 *** | $-5.502 \times 10^5$ *** | −0.001 *** | $-8.415 \times 10^5$ *** | −0.003 *** | $-3.075 \times 10^5$ ** | −0.001 *** | 0.000 *** |
| | (0.000) | (0.000) | (0.000) | (0.000) | (−0.001) | (0.000) | (0.000) | (0.000) |
| Residential Mobility (% population that moved within last year) | 0.000 | $-2.605 \times 10^5$ ** | $4.019 \times 10^5$ | $-3.325 \times 10^5$ * | (0.001) | −1.957 ** | $-6.231 \times 10^5$ | $-5.701 \times 10^5$ |
| | (0.000) | (0.000) | (0.000) | (0.000) | (0.000) | (0.000) | (0.000) | (0.000) |
| % of Population with No Health Insurance | 0.001 | 0.000 *** | 0.001 *** | $1.364 \times 10^5$ | $2.492 \times 10^5$ | $-1.090 \times 10^5$ | 0.001 | 0.000 |
| | (0.001) | (0.000) | (0.000) | (0.000) | (0.002) | (0.000) | (0.001) | (0.000) |
| COVID Year (yes = 1, no =0) | 20.598 ** | 1.024 ** | 1.643 | 0.132 | 19.601 | 0.983 *** | 9.411 | 2.353 |
| | (9.443) | (0.428) | (4.060) | (0.760) | (17.770) | (0.352) | (6.324) | (3.381) |
| Constant | 206.737 *** | 4.437 *** | 70.752 *** | 10.387 *** | 514.121 *** | 4.444 *** | 173.362 *** | 65.447 *** |
| | (26.731) | (1.213) | (11.492) | (2.150) | (50.301) | (0.996) | (17.902) | (9.571) |

Notes: * $p < 0.10$, ** $p < 0.05$, *** $p < 0.01$.

**Table 3.** Ordinary least squares regression of arrest rates for White and Black populations in Indiana using COVID-19 death rate.

| | Black | | | | White | | | |
|---|---|---|---|---|---|---|---|---|
| | **Violence** | **Homicide** | **Assault** | **Burglary** | **Violence** | **Homicide** | **Assault** | **Burglary** |
| | *b* (SE) | *b* (SE) | *b* (SE) | *b* (SE) | *b* (SE) | *b* (SE) | *b* (SE) | *b* (SE) |
| Some College | 0.002 * | 0.000 *** | −0.001 ** | 0.000 | −0.001 | $6.885 \times 10^5$ ** | −0.002 *** | −0.001 * |
| | (0.001) | (0.000) | (0.000) | (0.000) | (0.002) | (0.000) | (0.001) | (0.000) |
| Unemployment | −0.002 | 0.000 | 0.006 *** | 0.001 ** | 0.018 ** | 0.000 | 0.013 *** | 0.005 *** |
| | (0.004) | (0.000) | (0.002) | (0.000) | (0.008) | (0.000) | (0.003) | (0.001) |
| Average Household Income (measured in $10,000 units) | −0.002 *** | $-5.341 \times 10^5$ *** | −0.001 *** | $-8.442 \times 10^5$ *** | −0.003 *** | $-2.926 \times 10^5$ ** | −0.001 *** | 0.000 *** |
| | (0.000) | (0.000) | (0.000) | (0.000) | (0.001) | (0.000) | (0.000) | (0.000) |
| Residential Mobility (% population that moved within last year) | $-9.389 \times 10^5$ | $-2.384 \times 10^5$ ** | $3.607 \times 10^5$ | $-3.393 \times 10^5$ * | (0.000) | −1.781 ** | $-7.682 \times 10^5$ | $-6.069 \times 10^5$ |
| | (0.000) | (0.000) | (0.000) | (0.000) | (0.000) | (0.000) | (0.000) | (0.000) |
| % of Population with No Health Insurance | 0.001 | 0.000 *** | 0.001 *** | $1.727 \times 10^5$ | $-4.218 \times 10^5$ | $-1.338 \times 10^5$ | 0.001 | 0.000 |
| | (0.001) | (0.000) | (0.000) | (0.000) | (0.002) | (0.000) | (0.001) | (0.000) |
| COVID-19 Death Rate | 0.117 * | 0.005 ** | −0.006 | 0.001 | 0.110 | 0.006 ** | −0.065 | −0.016 |
| | (0.063) | (0.003) | (0.027) | (0.005) | (0.117) | (0.002) | (0.042) | (0.22) |
| Constant | 205.021 *** | 4.415 *** | 70.652 *** | 10.278 *** | 512.585 *** | 4.340 *** | 174.913 *** | 65.882 *** |
| | (27.040) | (1.234) | (11.574) | (2.164) | (50.702) | (1.007) | (18.005) | (9.633) |

Notes: * $p < 0.10$, ** $p < 0.05$, *** $p < 0.01$.

## 5. Discussion

The pandemic affected everyone in some way or another. The devastating mortality of over 1.1 million deaths and even greater cases of illness that occurred between 2020 and 2021 highlights a situation that is only seen by society once every century or so [1]. As the pandemic reshaped daily life across U.S. communities, this presents a unique opportunity to examine how such an impactful social change may shape behaviors like criminality. Although more data and research are needed to fully understand these relationships, early findings suggest that the pandemic had notable impacts on crime. Decreases in certain property crimes, alongside stability or increases in violent crime, highlight how the pandemic may have shifted the targets of crime and the strain faced by certain populations. While COVID-19 was felt by everyone, the current findings also suggest that this effect may have differed somewhat by race.

Our first research question asked how COVID-19 impacted Black and White arrests, net of controls for disadvantage, residential mobility, and education. Our findings indicated that COVID-19 had a significant connection with arrest rates for Black and White homicide and Black violent crime. This is an interesting finding as it suggests positive associations between COVID-19 and violence. Our second research question asked whether these effects differed across offense type. The results showed that both Black and White homicide arrests significantly increased following the onset of COVID-19. However, we acknowledge that the results from Indiana may not reflect those in other, more diverse states. Indiana is a predominantly White state (84%), and the results may differ in other locations that have different racial/ethnic compositions (a point we return to below) [30].

Our last research question asked whether the effects of COVID-19 on crime were felt more by some race groups than others. Arrest rates increased for the general Black violent crime category, but not for Whites, implying that COVID-19 may have shaped Black arrests to a greater degree. These findings suggest that there were some differences in COVID-19–crime relationships across race. However, questions remain about whether COVID-19 affected arrests in "racially invariant" ways, and we are unable to discern the precise reason for these different effects using the current data. Prior research on policing, race, and crime suggests that it could be due to several factors. When arrests increase for one race and not another, it is often asked if this is due to shifts in offending or to changes in enforcement responses. As mentioned in Dunbar and Jones's (2021) work, New York saw a disproportionate level of Black residents being targeted and arrested for public health order violations. It is possible that the increase in violent crime arrests was simply due to increased policing or enforcement in Black communities for these offenses. It is important to note that during the summer months of 2020, riots across the United States were under way in response to George Floyd's death, which also could have impacted arrest statistics as some police departments responded to the protests [31]. However, changes in arrests resulting from these protests would be reflected in low-level offenses, and even then, most charges against protesters ended up being dropped [32]. Thus, it is unlikely that arrests resulting from the protests would dramatically skew annual data, especially for violent crimes including homicide.

However, it is also possible that the pandemic affected actual crime patterns in certain communities more than others. The strains from COVID-19 may have compounded the disadvantages already faced by many Black communities. As strain arguments suggest, those with fewer resources likely suffered more acutely during the pandemic. Economic opportunities and strains existed for many, but the push toward crime may have become even more compounded and pronounced for already disadvantaged Black communities. Based on the results from this analysis, there are some similarities in COVID-19's ties to both Black and White arrests, but there are also key differences that suggest that there is not complete "racial invariance" in COVID-19–crime relationships. The effects of COVID-19 (and several other controls) on arrest rates for violent crime differed across race. Thus, already disadvantaged communities may have seen greater impacts from the COVID-19 pandemic, resulting in greater arrests for violence.

*Limitations and Future Work*

Several important limitations need to be discussed regarding this study. Data availability was a key limitation, as most states did not have crime data available for 2021 (and many did not have 2020 data available). As mentioned previously, the UCR had limited official data posted past 2019. As such, we relied on data from Indiana, which offered multiple years of arrest data covering the onset of the pandemic, disaggregated by race/ethnicity for multiple offense types. However, this limits generalizability, as Indiana is 84% White [30], and the nation as a whole is only 54% non-Hispanic White. Using a more diverse state for replication and further analysis would be useful moving forward.

There is also a need for additional studies that incorporate a wider range of variables and controls to account for the other aspects of social life that shifted during the pandemic, such as mental health, inflation, other opportunities for crime, and additional employment metrics (among others). There are other important social factors and changes that may have occurred but which we could not capture with the census data used here. Another limitation was the cross-sectional nature of our models. There is a need for time series analyses to see how changes in predictors were related to changes in crime rates throughout the pandemic, as this is a more powerful and robust analytic approach.

One final limitation is the caveat of using official police data. While COVID-19 may have caused changes in behavior among people, there is always the possibility that these data reflect changes in policy and enforcement rather than actual changes in crime. During the pandemic, social gatherings were limited, people began working from home, and city streets were seeing less activity. Thus, when violent crime did occur, it may have been more easily identified or enforced by police. In addition, this may have been more common in minority communities than in White communities. More research is needed to determine if crime spikes during COVID-19 (especially for less serious offenses that allow for more discretion, and for minority groups) were due to changes in behavior or enforcement. Using triangulation with other databases that do not rely on official statistics would help in this process.

Further work is needed to obtain a more complete understanding of COVID-19's effect on crime across race. This study offers an initial look at these relationships across Indiana counties. The findings presented here suggest that arrests for violence (but not burglary) rose in response to the pandemic for both Black and White populations. However, we have also found some indications that Black arrests for violent offenses were more consistently and positively tied to COVID-19 measures. Although additional research is needed to more fully understand how racial patterns in arrests changed in response to the pandemic, the current study offers important initial steps toward this goal.

**Author Contributions:** Conceptualization, C.P.; methodology, C.P. and B.F.; software, SPSS version 29. 0.0.0.; validation, C.P. and B.F.; formal analysis, C.P.; investigation, B.F.; resources, C.P. and B.F.; data curation, C.P.; writing—original draft preparation, C.P.; writing—review and editing, C.P. and B.F.; visualization, C.P. and B.F.; supervision, B.F.; project administration; B.F. funding acquisition, B.F. All authors have read and agreed to the published version of the manuscript.

**Funding:** This research was funded by the National Science Foundation under Grant No. 1849209. Any opinions, findings, and conclusions or recommendations expressed in this material are those of the author(s) and do not necessarily reflect the views of the National Science Foundation. The APC was funded by MDPI Societies Editorial Office.

**Institutional Review Board Statement:** Not applicable.

**Informed Consent Statement:** Not applicable.

**Data Availability Statement:** The data that supported the findings of this study are openly available in the Indiana State Police database (MPH) at https://www.in.gov/mph/projects/arrests-dashboard/ (accessed on 15 October 2022).

**Conflicts of Interest:** The authors declare no conflicts of interest.

## Notes

[1]    Data were fully available for 26 counties in Indiana. These 26 counties are the most populous in Indiana and had the most reported crime data. All of the excluded counties have low populations and few arrests (some as low as 70 arrests for the entire year) and even fewer violent crime arrests (some as low as 17 arrests for the entire year). Thus, the counties examined here offer the most reliable data and are less susceptible to spikes or yearly fluctuations that could dramatically alter results. We acknowledge that limiting the analysis to 26 counties is a caveat. However, the supplemental analysis that included additional counties with mean replacement for missing data provided similar findings (but with less reliable results across models).

[2]    Multicollinearity checks were performed among the control variables to ensure that correlations among predictors were not an issue. Correlations were all below r = 0.876 and VIF scores were all below standard cutoffs (below 4.0).

[3]    A count-based model (negative binomial) was run to ensure that the results were not biased due to the OLS regression assuming a normal distribution. The results were similar, indicating no bias was found.

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
