# Peer review of "COVID-19, Race, and Crime: An Early Look at Racial Disparities in U.S. Arrest Data throughout the COVID-19 Pandemic"

_societies, doi:10.3390/soc14030037_

Round 1

Reviewer 1 Report

Comments and Suggestions for Authors

Just a few structure and editing suggestions:

1. Figures 1-3 could be moved in the text to ensure a better flow of information and location of the figures closer to the relevant text.

2. I'm not a stats expert but the data in Tables 2 and 3 Average Household income through No Health Insurance isn't particularly accessible. Could these figures be presented in a different way? (No is an OK answer but it's worth thinking about accessibility here in relation to potential future citations and research use).

3. Although George Floyd and the resultant protests is mentioned on p13, This could have been a significant confounding factor and more reflection on this may be useful.

Reviewer 2 Report

Comments and Suggestions for Authors

the work is well written; the analysis of recent literature, including documentary sources and official statistics, is described clearly and with an adequate level of detail.

The theoretical framework is balanced compared to the other parts of the article.

The notes must be revised so as not to leave only the numbers at the foot of the page and the way of reporting the bibliographical references. From the editorial rules of the Journal: In the text, reference numbers should be placed in square brackets [ ], and placed before the punctuation; for example [1], [1–3] or [1,3]. For embedded citations in the text with pagination, use both parentheses and brackets to indicate the reference number and page numbers; for example [5] (p. 10). or [6] (pp. 101–105).

The reference list should include the full title, as recommended by the ACS style guide. Style files for Endnote and Zotero are available.

Reviewer 3 Report

Comments and Suggestions for Authors

Dear author(s), 

your manuscript is very well structured, useful and I find it very interesting. It is very rare, that I do not have ideas how to improve this text. I havo no remarks. I wish your manuscript many quotations.  

Author Response

Thank you for your time and consideration in reviewing this work. We thank you for the kind words and we are glad you believe it has merit for this journal.